# Fabrication of Refractory Materials from Coal Fly Ash, Commercially Purified Kaolin, and Alumina Powders

**DOI:** 10.3390/ma13153406

**Published:** 2020-08-02

**Authors:** Saidu Kamara, Wei Wang, Chaoqian Ai

**Affiliations:** 1Department of Chemical Engineering, School of Water Resources and Environment, Chang’an University, Xi’an 710054, China; ksaidu2013@gmail.com (S.K.); acq0003@live.com (C.A.); 2Key Laboratory of Subsurface Hydrology and Ecology in Arid Areas, Ministry of Education, Chang’an University, Xi’an 710054, China

**Keywords:** experiment, fabrication, coal fly ash, kaolin, refractory, sinter, sample

## Abstract

Coal fly ash and kaolin are ceramic compounds utilized as raw materials in the production of refractories. Fly ash is an environmental pollutant that emanates abundantly from coal thermal power plants. The management of the large amounts of fly ash produced has been very challenging, with serious economic and environmental consequences. Kaolin, on the other hand, is a natural and synthetic clay material used in medicines, paper, plastics, and cosmetic preparations. In this research, refractory materials (cordierite (Mg_2_Al_4_Si_5_O_18_), mullite (3Al_2_O_3_·2SiO_3_), and kyanite (Al_2_SiO_5_)) were fabricated in four different experiments, and an assessment was made of the strength of each of the materials. Coal fly ash and kaolin were each blended with alumina as starting materials. MgO and AlF_3_·3H_2_O were each applied as additives to the reacting materials. The mixtures were molded and sintered at temperatures between 1000 °C and 1200 °C for three hours in a muffle furnace, and characterized by SEM and XRD. The analysis revealed the evolution of cordierite, mullite, and kyanite alongside other crystalline compounds. The formation of kyanite in experiment C, due to the addition of AlF_3_·3H_2_O, is unprecedented and phenomenal. The XRD figures show the corundum phases crystallize at 1100 °C in experiments A and B, and disappear at 1200 °C.

## 1. Introduction

Mullite, cordierite, and kyanite materials are well-known compounds made of magnesium, aluminum, and silicate oxides. They have unique properties with advantages for a wide range of applications in ceramic and refractory industries [1,2,3,4,5,6]. These properties include excellent thermal shock resistance, sensitive creep resistance, high melting point, high mechanical strength, low thermal expansion, good chemical resistance, excellent electrical resistivity, resistance to abrasion, and is insoluble in hydrogen fluoride [7,8,9,10,11,12,13,14,15]. Refractories are used to produce microelectronic components, catalytic substrates, firing trays in furnaces, metal filters, catalyst carriers in automobile exhausts, gas turbine heat exchangers, porous ceramics, and circuit boards [7,8,11,16,17,18,19,20]. Refractories are marketable in chemical, petrochemical, industrial heating processes, power generation, and heavy industrial processing industries.

As mullite is not commercially available, it is prepared synthetically [11]. Refractory materials are fabricated in the laboratory, through the two most commonly used methods, solid-state reaction and wet chemical (e.g., sol-gel process, hydrolysis, spray pyrolysis, and thermal synthesis) [2,7,21,22,23,24,25]. The raw materials for these methods are expensive and yield very low products and are therefore, not preferred for a large-scale, low-cost processes [4,8]. In recent times, such refractory materials have been synthesized in laboratories in commercial abundance, through solid-state sintering reaction with inexpensive raw materials [4,8,11]. Fly ash and kaolin are among the raw materials widely used in the production of refractory materials.

Research has proven the nature of additives such as MgO in the fabrication of refractory materials and promoted their utilization in various applications [26]. The use of MgO and AlF_3_·3H_2_O, with fly ash as a raw material, in the production of refractories is promising considering their easy availability and production costs. Fly ash is a powdered raw material that can be directly mixed as a paste in ceramic production [26]. Studies have reported the applications of several sources of Al and Si (rice husk ash, demolition wastes, blast furnaces, slag, volcanic ash, etc.) for the production of refractory materials [27,28,29,30,31]. Fly ash is a cheap, darkly shaded, and fine powdered material with variable particle size distributions and chemical and mineralogical compositions, which depend on the quality and the temperature of the coal [29]. Commercial kaolin, on the other hand, is light shaded and more expensive than fly ash, but has a similar composition and particle size distribution [29,32]. It has been reported by various researchers that fly ash can be utilized in different structures, such as in roads and railways as a subgrade, railways construction, and landfill liners [33].

This paper is a report of four comparative experiments of different reacting materials and additives, leading to the fabrication of three different refractory crystalline materials. Coal fly ash, purified commercial kaolin {Al_2_Si_2_O_5_(OH)_4_}, and alumina (Al_2_O_3_) are the starting materials in these experiments using magnesium oxide (MgO) and aluminum fluoride trihydrate (AlF_3_·3H_2_O) as additives. Fly ash is a residue generated from the combustion of powdered coal in thermal power generation industries [13,34]. Coal is the primary source of energy in China, which produces an enormous quantity of fly ash leading to severe environmental issues, with increasing public attention [13,35,36]. Coal fly ash is currently a top source of industrial solid waste in China, which in severity pollutes the air, soil, and water [36,37]. This eventually affects the food chain and poses a threat to the health of inhabitants. Fly ash production is increasing terribly. The increasing volume of fly ash generated in power industries is getting expensive to manage and difficult to dispose of, and therefore using it in various applications, such as engineering projects, is the most favorable method for the management of this type of waste [33]. The focus was, therefore, to fabricate refractory materials from fly ash, and make contrasting analyses with those fabricated from kaolin. The various refractory crystallines from fly ash and kaolin were identified, and their porosity was physically examined. Various researchers have reported that the porosity of the materials contributes to the mechanical strength and density of refractory material. The properties of fly ash are suitable for use in refractory and ceramic industries [1], and can thus be used as an alternative lower-cost material to alumina. Kaolin is a mineral extensively used in the medical and engineering industries [38,39]. Coal fly ash and kaolin are super refractory compounds whose refractory products can withstand heat without melting. The particles in a refractory compound bind together when they are fired at relatively low temperatures, and the bonding process that occurs is called sintering. Sintering in a ceramic correlation is the bonding of particles with an increase in temperature. The material is sufficiently fired so that it does not break when it comes into contact with water. Crystallization to solid-state is achieved by subjecting silica-alumina to high temperatures. The effect and use of additives in the synthesis of refractory materials is to improve the bonding and densification of the reacting materials at various temperatures. The chemical purity and particle size distribution of Al_2_O_3_ and SiO_2_ precursors play a significant role in the temperature, and rate, of mullite formation. It has been reported that high mullite refractories may be achieved by firing natural kyanite group minerals as raw materials of anhydrous aluminum silicate polymorphs, andalusite, kyanite, and sillimanite with the same composition (Al_2_SiO_5_ or Al_2_OSiO_4_) [3]. Some common raw materials containing Al_2_O_3_ and SiO_2_ are commercially available powders of alumina and silica, bauxite, and sillimanite minerals.

The aim was to fabricate refractory composites from the powders of coal fly ash (a pollutant from coal thermal plants) and kaolin using two different additives (MgO and AlF_3_·3H_2_O) with varying stoichiometric parameters at varying temperatures, and to qualitatively examine their porosities, in order to determine the strength and insulating properties of the materials fabricated from each of the raw materials.

## 2. Materials and Experimental Procedures

### 2.1. Materials

Alumina (Al_2_O_3_), Fly Ash (SiO_2_), Kaolin {Al_2_Si_2_O_5_(OH)_4_}, Magnesium Oxide (MgO), and Aluminum Fluoride Trihydrate (AlF_3_·3H_2_O) (Table 1). The coal fly ash (SiO_2_ ˃ 99%) was bought from vendors in Xi’an Linyuan Silica Limited, Xi’an, China, while alumina, kaolin, magnesium oxide, and aluminum fluoride trihydrate are stored laboratory research chemicals. Glassy SiO_2_ is the most abundant mineral phase in coal fly ash. The other mineral phases are hematite, magnetite, quartz, feldspar, corundum, and anhydrite.

### 2.2. Instruments

Sartorius analytical electronic balance (Beijing Zhonghuan Instrument Equipment Co. Ltd., Beijing, China), weighing papers (Bio Life Science and Technology Co. Ltd., Shanghai, China), planetary ball grinding mill (Liyou Machinery Equipment Co. Ltd., Xi’an, China), E-ZIRCON hydraulic press jack (Danyang Ouba Co. Ltd., Danyang, China), electrothermal constant temperature dry box (Shanghai East Star Building Materials Testing Equipment, Shanghai, China), muffle furnace -1300 °C maximum heating capacity (Tianjin Zhonghuan Experimental Furnace Co. Ltd., Tianjin, China), X-ray diffraction (XRD) (Bruker, Karlsruhe, Germany), scanning electron microscopy (SEM) (S-4800, Hitachi, Tokyo, Japan), and jade 6 and origin 9 software.

### 2.3. Material Processing and Procedures

Four experiments A, B, C, and D were demonstrated, leading to the fabrication of cordierite, mullite, and kyanite composite refractories. Alumina (Al_2_O_3_), coal fly ash (SiO_2_), and kaolin {Al_2_Si_2_O_5_(OH)_4_} in Table 1 above were the raw materials used in this research. The three materials were processed and obtained A (10 g, 4 g), B (10.2 g, 12.9 g), C (10 g, 4 g), and D (10.2 g, 12.9 g) stoichiometric values using an electronic weighing scale. Experiment A was demonstrated by blending 10 g of Al_2_O_3_ g and 4 g of SiO_2_, adding 1 g of MgO to enhance the process. In experiment B, 10.2 g of Al_2_O_3_ and 12.9 g of Al_2_Si_2_O_5_(OH)_4_ were combined using 2 g of magnesium oxide (MgO) as an additive to the reactants. Experiment C was conducted using the same raw materials and stoichiometric values as experiment A, but 1g of aluminum fluoride trihydrate (AlF_3_·3H_2_O) was used as an additive. In addition, experiment D was performed using the same reacting materials used in experiment B but 2 g of AlF_3_·3H_2_O was added.

Three samples, labeled a, b, and c in each of the four experiments, were prepared, and each was weighed and subsequently placed in a grinding mill for 10 min for further mixing. This was followed by molding the samples into tablet-shaped materials using an EZIRCON hydraulic press jack to make them ready for firing. The samples were sintered at 1000 °C, 1100 °C, and 1200 °C in a muffle furnace for 3 h at a constant temperature rate of 10 °C per minute. The maximum heating capacity of the muffle furnace used in this experiment was 1300 °C. The three sets of samples were granulated and ground for SEM and XRD respectively. This procedure was done for all specimens in experiments A, B, C, and D.

Characterization was carried out using SEM and XRD techniques. Using Jade and Origin software, the data obtained from the XRD characterization were analyzed to identify and confirm the formation of refractory compounds from the sintered samples. SEM analysis was done to study the changes in microstructure and morphology during the sintering of the specimens, while XRD provided information on the crystalline phase composition in the specimens.

Owing to the lack of electrical conductivity in ceramics, the samples before the SEM characterization were subjected to a gold electric conductor for 10 min, to enhance the conductivity of the samples. This was subsequently followed by placing the samples in a scanning electron microscope for 10 min, with the purpose of determining the surface morphology.

The SEM (S-4800, Hitachi, Tokyo, Japan) equipment was operated at a voltage of 10 Kv to determine the morphology of the sintered materials. The XRD (Bruker, Karlsruhe, Germany), equipment with specifications AXS D8 with CuK-α, was operated at 40 mA. The XRD peaks were recorded between a 10° and 90° range of angles (2Ɵ) at a scanning rate of 15°/s. Particles with irregular shapes were shown by the SEM analysis for both samples in experiments A, B, C, and D.

## 3. Results and Discussion

### 3.1. Scanning Electron Microscope

Figure 1 below shows SEM analysis of the samples in experiment A. The morphology of the refractory materials shown in Figure 1 was influenced by the amount of content it contains. The particles observed in the material are prismatic, orthorhombic, and dense in structure. The layered structures are interconnected with each other with no visible pores. The highly dense crystalline material is an indication of the presence of little or no contaminants with particle sizes significant enough to agglomerate into bulky substances during the sintering process. The connected or closed micrographic crystals significantly contributed to the low water absorption of the sintered material.

The micrographs in Figure 2 below show no well-defined geometrical planes. It is observed from the morphology of the SEM images that the crystal structures of the samples were highly dense and non-porous. The porosity of a sample normally affects its mechanical property, which is a significant factor that determines the bending strength. The non-porosity observed in the above micrographs renders the refractory materials with the possibility of high density, high strength, and resistance to corrosion.

The particles shown in the SEM images of Figure 3 below are geometrical irregular in shape, with no visible pores between the crystal lattice structures. However, the grain sizes of the samples (b_1_, b_2_) are significantly different from those observed in samples a_1,_ a_2_ and c_1_, c_2_. The agglomerate layered crystals are seen to be rectangular, prismatic, and orthorhombic shaped, except for samples b_1_, b_2_, which have little defined crystalline structures.

The application of AlF_3_·3H_2_O as an additive to the alumina and fly ash as the reacting materials in experiment C resulted in the evolution of non-porous geometric crystalline materials. This is similarly seen in the crystalline structures produced in experiment A, when MgO was added to alumina and fly ash reacting materials. However, the result obtained from the XRD shows kyanite as the main refractory material produced in C, while cordierite and mullite are the main refractory products in A. The refractory materials produced in both cases exhibit the property of having good mechanical strength due to the non-porosity of the crystalline material [40]. The possible chemical reaction for the formation of kyanite is ${\mathrm{Al}}_{2}O_{3}+{\mathrm{SiO}}_{2}\overset{{\mathrm{AlF}}_{3}\cdot 3H_{2}O}{\to }{\mathrm{Al}}_{2}{\mathrm{SiO}}_{5}$.

The micrographs shown in Figure 4 below have no well-defined geometrical crystalline shapes. The crystalline structure of the sample sintered at 1000 °C is non-porous and highly dense. Porosity is pronounced in the samples sintered at 1100 °C and 1200 °C, but the pore sizes are different between the two samples. A highly porous refractory material has excellent insulating properties. Some of the crystalline structures of the sample sintered at 1100 °C appear to be bulky and slightly separated from each with large pore sizes. The pore size affects the thermal conductivity, water absorption, and the mechanical strength of refractory materials.

It should be noted that the addition of AlF_3_·3H_2_O as an additive to the starting materials (alumina and kaolin) in experiment D resulted in the formation of crystalline materials with significant porosities at 1100 °C. The porosity was so high because some of the crystalline materials were probably not able to undergo bonding during the sintering. The temperature, grain size, particle distribution, composition of the materials, and the sintering vicinity are important factors that affect the diffusion of the grain boundary and the volume of the materials. Sintering causes the crystalline materials to bond together, resulting in densification and reduction of porosity. The results obtained from the XRD analysis also show an increase in the number of crystalline phases formed in the samples. On the other hand, the use of MgO as an additive to alumina and kaolin as starting materials in experiment B, resulted in the evolution of non-porous crystalline structures, which is a significant functional property for the mechanical properties of the refractory materials formed.

### 3.2. X-ray Diffraction

Information about the crystal structure of the sintered material was obtained using an x-ray diffraction technique. The samples were scanned between a 10° to 90° range of angles at a voltage of 40 Kv and a maximum intensity of 1272. The figures below show the peak patterns of the samples in experiments A, B, C, and D sintered at 1000°, 1100°, and 1200°. Analysis indicates the formation of cordierite, mullite, and cordierite composites in both experiments.

XRD was used to identify the crystalline phases of the material and reveal information about its chemical composition, by comparing the obtained data to that in the reference databases. The XRD technique is a non-destructive test, and it is an important tool for evaluating minerals, polymers, corrosion products, and unknown materials using finely-ground powdered prepared samples. The technique is performed by targeting an x-ray beam at a sample and measuring the scattered intensity as a function of the outgoing direction. Once the beam is separated, the scatter, also called a diffraction pattern, indicates the crystalline structure of the sample.

#### 3.2.1. Analysis for Experiment A

According to Figure 5 below, characteristic cordierite peaks were depicted at 2ϴ = 21.6°, 26.9°, and 28.3°. The transformation to a mullite phase for samples a, b, and c began at 2ϴ = 30.9°, 25.9°, and 26.9°. The main crystalline phases at 1000 °C were cordierite, mullite, alumina, and quartz. The XRD analysis shows an unprecedented number of corundum crystalline phases with high intensities when the temperature was increased from 1000 °C to 1100 °C. Cordierite, mullite, and corundum phases were formed at 1100 °C. The results from the analysis further show a total disappearance of corundum crystalline phases when the temperature of the sample was increased to 1200 °C. Various researchers have reported that the addition of MgO increases the strength of refractory products, giving it additional advantages for its use in engineering and refractory industries.

#### 3.2.2. Analysis of Experiment B

Figure 6 below shows the XRD patterns of five crystalline structures. According to results from the analysis, mullite and cordierite refractories were formed alongside the crystals of cristobalite, quartz, and corundum. A significant number of corundum crystalline phases were formed at 1100 °C. An increase in temperature to 1200 °C also resulted in the disappearance of the corundum crystals. The transformation pattern of the corundum was similar to those in experiment A. The formation of cordierite in all three samples began at 2ϴ = 21.6° while the mullite phase began at 2ϴ = 25.9° and 2ϴ = 16.3°.

Experiments A and B had different starting materials, with different stoichiometric measurements and additives, but their results interestingly show a similar pattern of behavior for samples sintered at 1000 °C, 1100 °C, and 1200 °C, and in both cases the corundum crystalline phases disappeared as the heating temperature was increased from 1100 °C to 1200 °C.

#### 3.2.3. Analysis of Experiment C

Figure 7 below shows the phases of the seven crystalline structures formed (cordierite, mullite, corundum, alumina, quartz, kyanite, and moganite) in the three sintered samples. It can be seen from the XRD patterns below that kyanite was the major refractory material formed in this experiment. The evolution of kyanite begins at 2ϴ = 20.6°, and its formation occurs along the same plane of 2ϴ in the three samples. Kyanite formation in this experiment was unprecedented and phenomenal, and this shows a marked difference to the refractory materials produced in experiments A, B, and D, in which mullite and cordierite are the main refractory products. Studies have shown that kyanite can be converted to mullite when calcined above 1250 °C.

#### 3.2.4. Analysis of Experiment D

The result from the XRD analysis in Figure 8 shows the evolution of seven crystalline phases in the specimens. According to the analysis, cordierite and mullite were the two refractories products formed in all the three samples. The formation of cordierite and mullite crystals started at 2ϴ = 21.6° and 2ϴ = 16.4°, respectively, and the pattern of formation of these refractory materials was observed along the same plane of 2ϴ in all the three samples.

Unlike the addition of MgO to the starting materials (alumina and kaolin) in experiment B, the unprecedented number of crystalline structures seen to occur across the various temperatures of the three samples in experiment D, can be linked to the addition of AlF_3_·3H_2_O as an additive to the starting materials.

## 4. Discussion

Cordierite, mullite, and kyanite were successfully fabricated from four different experiments A, B, C, and D, using fly ash and kaolin as the raw materials. Magnesium oxide and aluminum fluoride trihydrate were both utilized as additives to the reactions. The specimens were sintered at temperatures of 1000 °C, 1100 °C, and 1200 °C. The densities of the fabricated materials were very high and non-porous, according to the SEM micrographs. However, porosity was observed for samples in experiment D sintered at 1100 °C. The XRD results show significant crystalline materials formed across the four experiments.

## 5. Conclusions

SEM images in experiments A and C showed geometrical large sizes, non-porous, and compactly layered crystalline structures. This implies that the addition of both MgO and AlF_3_·3H_2_O in experiments A and C produced refractory materials with increased mechanical properties. However, the unprecedented fabrication of kyanite refractory compound in experiment C showed a significant difference from the refractory materials produced in experiment A. This difference in the fabricated refractory materials between experiments A and C was due to the application of AlF_3_·3H_2_O as an additive to alumina and fly ash reacting materials in experiment C. Therefore, adding AlF_3_·3H_2_O produces kyanite with very good refractory properties. Studies have proven that kyanite is a super-duty refractory material with very high resistance to thermal shock.

Furthermore, comparing the results in experiments B and D, the crystalline materials in B were irregular and highly dense with no visible pores, while images in experiment D showed pronounced porosity for the specimens sintered at 1100 °C and 1200 °C, with those at 1100 °C slightly separated from each other. The difference in the result of the materials in experiment D to those in B was due to the application of AlF_3_·3H_2_O as an additive to the reactants. Therefore, it can be deduced that adding MgO to alumina and kaolin in experiment B produced refractory materials with good mechanical properties and high densities, as opposed to adding AlF_3_·3H_2_O to alumina and kaolin in experiment D. In addition, it has also been reported from other studies that porous refractory materials have high insulating properties. In this research, three refractory materials were obtained from experiments involving fly ash and two from kaolin.

Fly ash is a tremendous environmental pollutant in the coal power industry and has been discovered to be a useful raw material in ceramic and refractory industries and could, therefore, be a preferred precursor to kaolin as a way to alleviate the threat it poses to the environment. Coal fly ash could be utilized to fabricate refractory compounds, primarily to preserve the environment, and also as a cost-effective material in the production of refractories.

## Figures and Tables

**Figure 1 materials-13-03406-f001:**
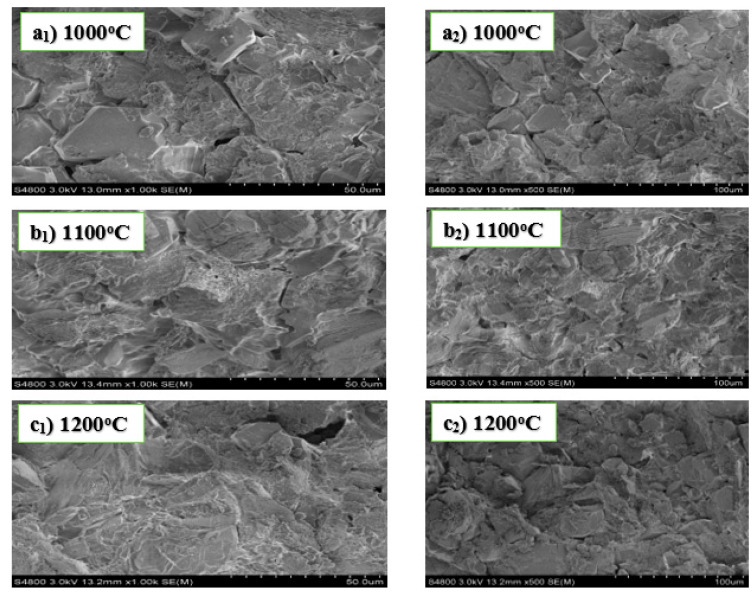
SEM images of experiment A sintered at 1000 °C (**a_1_**,**a_2_**), 1100 °C (**b_1_**,**b_2_**), and 1200 °C (**c_1_**,**c_2_**).

**Figure 2 materials-13-03406-f002:**
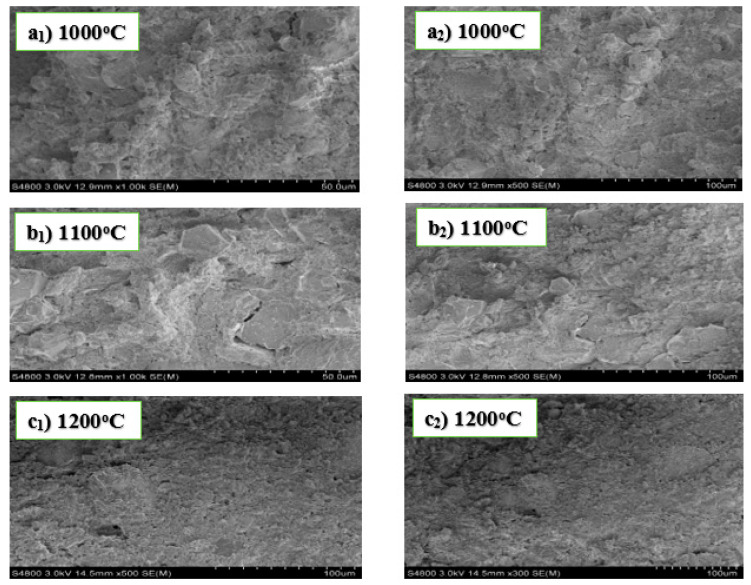
SEM images of experiment B sintered at 1000 °C (**a_1_**,**a_2_**), 1100 °C (**b_1_**,**b_2_**), and 1200 °C (**c_1_**,**c_2_**).

**Figure 3 materials-13-03406-f003:**
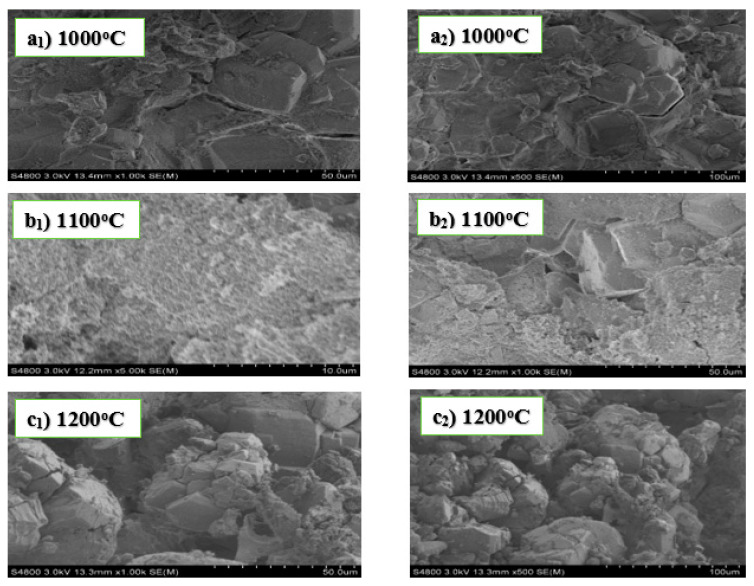
SEM images of experiment C sintered at 1000 °C (**a_1_**,**a_2_**), 1100 °C (**b_1_**,**b_2_**), and 1200 °C (**c_1_**,**c_2_**).

**Figure 4 materials-13-03406-f004:**
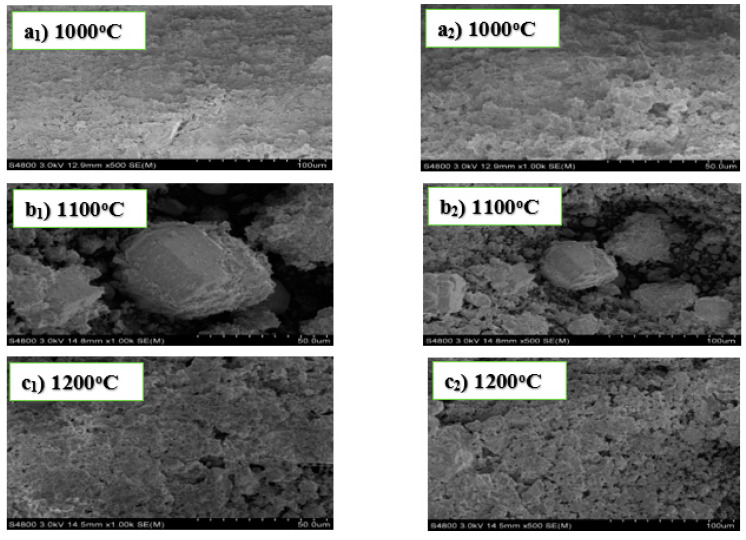
SEM images of experiment D sintered at 1000 °C (**a_1_**,**a_2_**), 1100 °C (**b_1_**,**b_2_**), and 1200 °C (**c_1_,c_2_**).

**Figure 5 materials-13-03406-f005:**
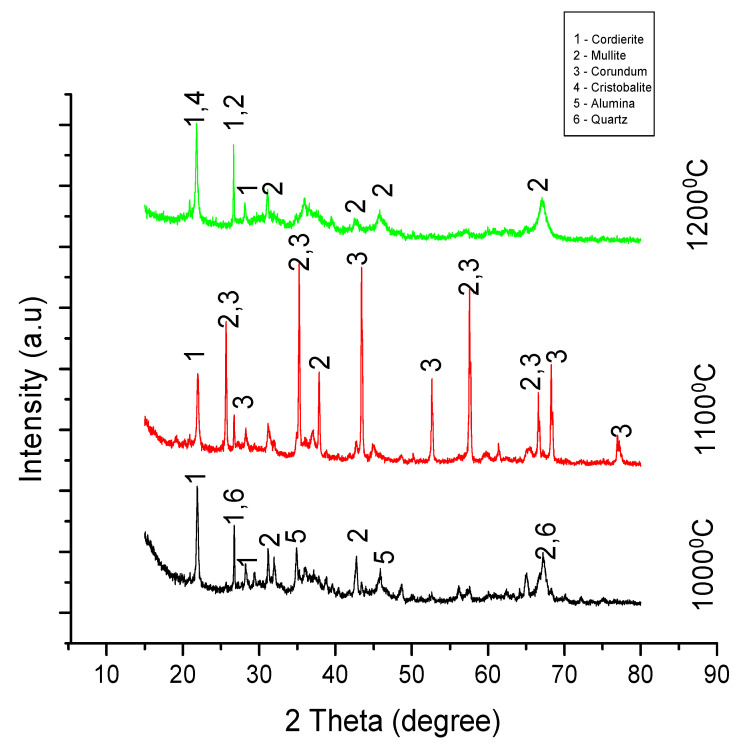
XRD peaks for samples in experiment A sintered at 1000 °C, 1100 °C, and 1200 °C.

**Figure 6 materials-13-03406-f006:**
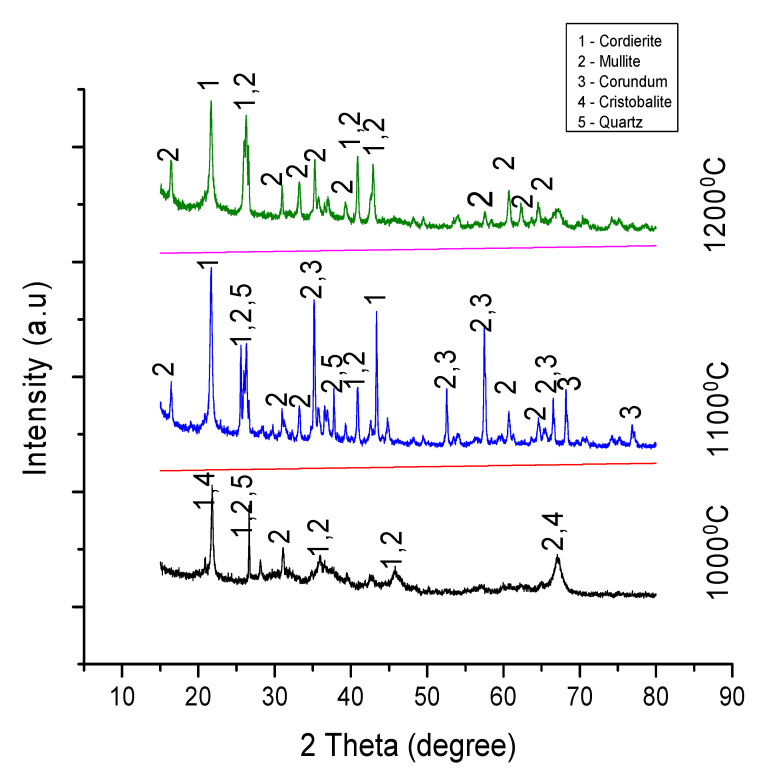
XRD peaks for samples in experiment B sintered at 1000 °C, 1100 °C, and 1200 °C.

**Figure 7 materials-13-03406-f007:**
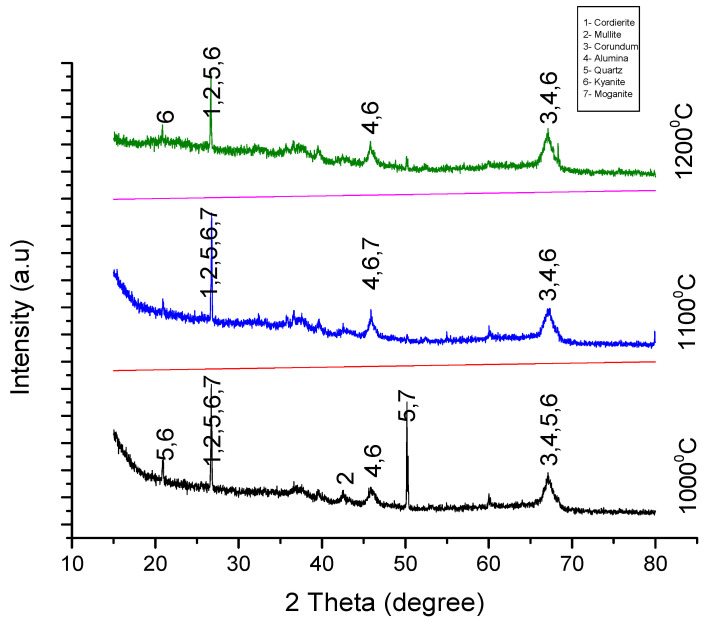
XRD Peaks for samples in experiment C sintered at 1000 °C, 1100 °C, and 1200 °C.

**Figure 8 materials-13-03406-f008:**
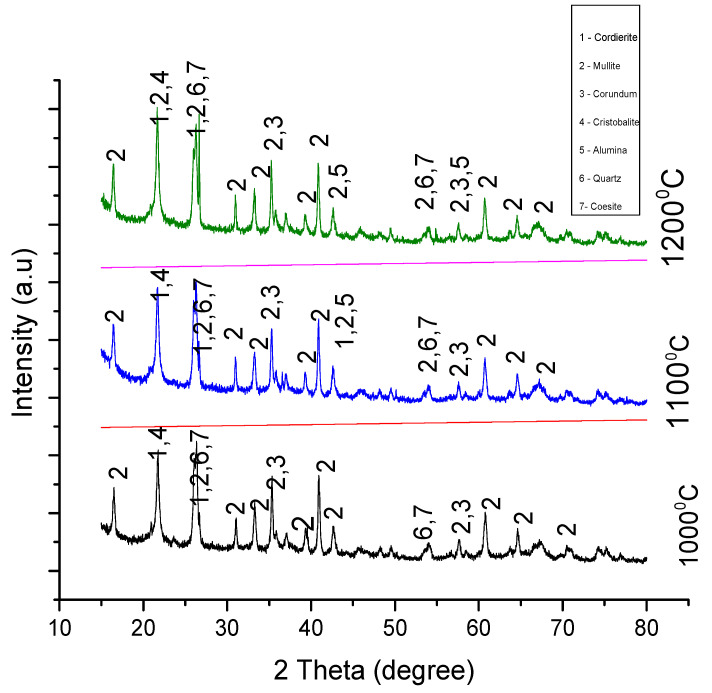
XRD peaks for samples in experiment D sintered at 1000 °C, 1100 °C, and 1200 °C.

**Table 1 materials-13-03406-t001:** Stoichiometric data of the reactants in experiments A, B, C, and D.

	Reacting Materials	Additives	Temperature (°C)		
			1	2	3
Experiment A	Al_2_O_3_ (10g) SiO_2_ (4g)	MgO (1g)	1000	1100	1200
Experiment B	Al_2_O_3_ (10.2g) Al_2_Si_2_O_5_(OH)_4_ (12.9g)	MgO (2g)	1000	1100	1200
Experiment C	Al_2_O_3_ (10g) SiO_2_ (4g)	AlF_3_·3H_2_O (1g)	1000	1100	1200
Experiment D	Al_2_O_3_ (10.2g) Al_2_Si_2_O_5_(OH)_4_ (12.9g)	AlF_3_·3H_2_O (2g)	1000	1100	1200

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
