# Peer review of "Fabrication of Refractory Materials from Coal Fly Ash, Commercially Purified Kaolin, and Alumina Powders"

_materials, 2020, doi:10.3390/ma13153406_

Round 1
Reviewer 1 Report
The authors state that samples containing Al2O3 (10g) + SiO2 (4g) + AlF3.3H2O) (1g) show good mechanical properties, on which basis such studies were not carried out, whether it is only a presumption or maybe such studies are available somewhere, if so, maybe it's worth supplementing the article with it. Similarly, in other cases, it is referred to as density, but it is probably assessed solely on the basis of SEM images.
It is also unclear why in the case of samples Al2O3 (10.2g) + Al2Si2O5 (OH) 4 (12.9g) + AlF3.3H2O (2g) at 1000 c the porosity is low and at 1100 C high, then again at 1200 C , this does not seem logical in the sintering process. What causes just at 1100 C the formation of pores, which later disappear, requires explanation.
Author Response
Dear Reviewer,
Please find attached response

Reviewer 2 Report
Dear authors,
Thank you for submitting your manuscript entitled "Fabrication of refractory materials from coal fly ash, commercially purified kaolin, and alumina powders" to MDPI Materials. This paper is a report of four comparative experiments of different reacting materials and additives leading to the fabrication of three different refractory crystalline materials.
This manuscript represents a scientifically rigorous study and qualifies for publication in MDPI Materials.
However, before publication, I suggest some minor revisions:
1) Use vector graphics throughout the manuscript
2) Add a brief discussion section before the conclusion section, as recommended by the journal guidelines.
3) Extend the literature review at the end of your introduction section.
Author Response
Dear Reviewer,
We have responded to the comments in the attachment.

Reviewer 3 Report
Dear Authors here some small remarks are outlined and after some small correction your article will be ready for publication. The positive merits of your article are shared with the Editor.
The authors quote too arbitrarily the statement from lit. source 11: „As mullite is not commercially available, it is prepared synthetically“Probably meaning that mullite for laboratory use. The authors of article wrote “Refractory compounds are not commercially common [11].”
The sentence: "Fly ash and kaolin are the most well-known solid materials for the production of refractory materials for the industrial production of refractory materials" needs correction, e.g. : Fly ash and kaolin are among the row materials widely used in production of refractories. The description of the HRD and CEM techniques should be moved to the experimental part.
The conclusions should be presented in a more generalized form, without the use of repetitions from the experimental part. The expression crystalline materials in B are non-geometrical requires correction.
More detailed comments:
Introduction:
1. To correct the statement: “Refractory compounds are not commercially common [11].”-line 36 because in [11] is written: „As mullite is not commercially available, it is prepared synthetically.“
2. The sentence: "Fly ash and kaolin are the most well-known solid materials for the production of refractory materials for the industrial production of refractory materials"-line 42 needs correction, e.g. : Fly ash and kaolin are among the row materials widely used in production of refractories.
Materials…:
1. To move the aim of the work at the end of the Introduction
Results:
1. To move description of the used XRD and SEM apparatuses in the Experimental part-line 127 and below
2. To exchange “non-geometrical shape” of particles with irregular –line 132 and below
Conclusion:
1. The conclusions should be presented in a more generalized form, without the use of repetitions from the experimental part.
Most of the remarks are connected with the arrangement of the work, hope following my advice the work will be more readable and more useful.
Author Response

(The authors gave the same response as above.)

Reviewer 4 Report
The manuscript Fabrication of refractory materials from coal fly ash, commercially purified kaolin, and alumina powders, presents an interesting study that attempts to investigate the production of fabricate composites of coal fly ash and kaolin by adding two different additives namely: MgO and AlF3·3H2O. The manuscript is well written. Figures and Tables are well presented. The results presented correspond to the research conclusions. However, there are a few corrections that must be made prior to the publication in the Journa: Materials. Therefore, Minor Revision is suggested.
Specific comments:
1). Line 88: Provide more specific details regarding the coal ash. Which is the mineralogical composition? 99% means that it is composed by either amorphous SiO2-glass or/and quartz. Which are the other accessory mineral phases?
2). Line 137-146: I think that these are general comments that do not add scientific value to the manuscript. I would suggest this part of the text to be omitted.
3). Lines 149, 167, 182, 213, 226, 241 and 253: Do not begin the section with the chemical formulas, without introducing the reader to the subject of the section. It is suggested either to make separate sections for each case or to integrate these chemical formulas within the manuscript.
4). Addition of AlF3·3H2O produces kyanite. Could you propose if possible a chemical reaction presenting the kyanite formation based on the literature?
5). The Figure numbering is quite confusing because the same letters are repeated e.g. a-a, b-b etc. Please modify the numbering by explaining that they correspond to pictures of separated samples.
6). Line 170: Please correct (c).
Author Response

(The authors gave the same response as above.)

Reviewer 5 Report
Dear Author,
The study is well planned and presented in an orderly manner. It needs further characterization of the synthesized compounds. But as an initial study it is well presented and can be accepted for publication in its current form.
Suggestions are:
- Further characterization needed to be done
- Applications needed to be tested.
- Systematic study of the experimental conditions should be carried out for complete optimisation.
- Correlation with the experimental conditions and the formation of the compounds should be explained.
Author Response
Dear Reviewer,
We have responded to the comments in the attachment
